# Patient-Reported Outcomes of Liposuction for Lipedema Treatment

**DOI:** 10.3390/healthcare11142020

**Published:** 2023-07-13

**Authors:** Fiona Kirstein, Matthias Hamatschek, Henning Knors, Marie-Luise Aitzetmueller-Klietz, Matthias Aitzetmueller-Klietz, Philipp Wiebringhaus, Charalampos Varnava, Tobias Hirsch, Maximilian Kueckelhaus

**Affiliations:** 1Department of Plastic, Reconstructive and Aesthetic Surgery, Hand Surgery, Fachklinik Hornheide, 48157 Muenster, Germany; 2Plastic and Reconstructive Surgery, Institute of Musculoskeletal Medicine, Westfalian Wilhelms-University, 48149 Muenster, Germany; 3Division of Plastic and Reconstructive Surgery, Department of Trauma, Hand and Reconstructive Surgery, University Hospital Muenster, 48149 Muenster, Germany

**Keywords:** lipedema, liposuction, QOL, PHQ-9, WHOQOL-BREF, restrictions in daily life

## Abstract

Background: Lipedema, as a disabling and consequential disease, is gaining more awareness due to its potential omnipresence. Patients suffering from lipedema show a characteristic painful display of symmetric accumulations of adipose tissue. The combination of swelling, pain and decreased quality of life (QOL) is outstanding for the diagnosis. The aim of this study was to identify the effect of liposuction in terms of the QOL for patients and underline important factors of current and pending research regarding surgical therapy of lipoedema. Methods: Patients suffering from lipedema prior to and after receiving liposuction at our hospital were included in this study. Patients completed a lipedema-specific self-designed 50 item questionnaire: the World Health Organization Quality of Life BREF (WHOQOL-BREF) and the Patient Health Questionnaire 9 (PHQ-9). A linear mixed model was used for outcome analysis. Results: In total, 511 patients completed a questionnaire prior to the surgery at primary presentation to the hospital and a total number of 56 patients completed a questionnaire after liposuction. A total of 34 of these patients filled in both questionnaires prior to and after surgery. The general characteristics of the disease, such as daily symptoms and psychological health, pertinently improved after surgery. Conclusions: Liposuction can have a general improving effect on the QOL of patients, both in private and professional life. Liposuction may currently be the most evident and promising method in the treatment of lipedema.

## 1. Introduction

Lipedema is a very common, progressive and often misdiagnosed disease that mainly affects women and leads to disabilities in performances in professional and private life [1,2]. Lipedema was first described by Allen et al. in 1940 and 1951 [2] and updated in 2012 by Herbst [3] (Table 1). During the past few years, awareness has risen and lipedema has been investigated more thoroughly [3,4,5].

Current research is suggesting that up to 15 percent of women suffer from lipedema [7], while many of them go undiagnosed or misdiagnosed [8,9]. The disease’s etiology remains poorly understood [6,10,11]. Clinically, the picture of lipedema displays painful symmetric adipose tissue accumulations in the early stages, especially affecting the lower extremities, while the upper extremities tend to be affected later [3,12]. The combination of swelling, pain and decreased QOL is outstanding for the diagnosis and must be distinguished from the misdiagnosis of adiposity, although concomitant obesity and increased body mass index (BMI) are common [13]. There are three stages of lipedema. Stage one describes a normal skin surface with enlarged hypodermis. Stage two is defined by an uneven skin with indentations in the fat and where larger mounds of tissue grow as unencapsulated masses, lipomas and angiolipomas. Stage three describes large extrusions of tissue causing deformations especially on the thighs and around the knees [3].

The consequences for lipedema patients include a negative impact on the QOL of patients with high levels of mental stress [14]. A diagnosis is mainly based on anamnesis and clinical examination, including inspection and palpation as well as ultrasound to rule out differential diagnoses such as chronic venous insufficiency or lymphedema [5,11,15,16]. Mostly, conservative treatments, such as complex decongestive therapy (CDT), are performed, including manual lymphatic drainage (MLD), intermittent pneumatic compression (IPC) and garments for compression [6,17]. These measures may show positive effects on appearance as well as pain levels [16,18].

Liposuction is especially considered when conservative therapy has failed [19]. Recently, there has been rising interest in this treatment option [20]. Previously, dry liposuction use was contraindicated due to causing lymphatic vessel damage [21]. Current methods of choice are micro-cannular tumescence anesthesia [22,23,24], water jet-assisted liposuction (WAL) [25] and power/vibration-assisted liposuction (PAL) [26], which are all suggested to be successful in terms of acute and long-term outcomes [27]. Early therapeutic intervention seems to be crucial for better results [28,29,30,31]. Several studies proposed a positive effect on quality of life, pain levels, professional performance and personal daily life [9,12,22,24,27]. Some studies suggested a concomitant lesser requirement of conservative treatment methods [23,31,32,33]. The aim of this study was to demonstrate the outcomes of liposuction on lipedema-associated symptoms such as pain levels, swelling or a feeling of tenderness as well as the effect of liposuction on patients’ QOL assessed with PHQ-9 and WHOQOL-BREF scores. Furthermore, the effect of liposuction on occupational disability was evaluated.

## 2. Materials and Methods

This study was approved by the local ethics committee. Code: 2021-684-f-S.

Patient Selection: Every patient who presented with a diagnosed stage of lipedema, according to the diagnostic criteria of Table 1. and who underwent liposuction for lipedema treatment in our hospital between December 2019 and December 2022 filled out an individually designed questionnaire (see Appendix A). Only patients who received one or more liposuctions and were aged over 18 years were included in the study. The median time between surgery and completion of the questionnaire was 3 months. Patients who came for primary presentation to the hospital prior to surgical intervention filled in a lipedema-specific questionnaire between May 2019 and May 2022. For this prior-to-surgery questionnaire, only patients who had not yet received liposuction and were of sufficient age were included to assess the quality of life of patients who did not yet receive surgical treatment for lipedema. The descriptive results of this questionnaire were separately described in a publication by Hamatschek et al. [14].

Questionnaire for post-liposuction: The results of the questionnaires prior to and after surgery were compared in this study. The QOL of lipedema patients at primary presentation was assessed and demonstrated a negative impact on daily life [14]. Those results were compared to the post-liposuction questionnaire (Appendix A) where general information such as height and weight before and after the surgery were monitored. Patients were asked to give information about the body sites that surgery was performed on, how many surgeries they had already had, if their weight changed and which body parts are still affected. Additionally, patients were asked if a compression garment was worn accordingly if manual lymphatic drainage was performed and how it affected their condition. Moreover, their limitations in professional life were assessed as well as their physical activities. Post-operative pain and swelling, circulatory problems and other complications were evaluated. Patients were also asked about their smoking habits and previous thrombosis. Further questions included whether they contacted a support group and obtained a second opinion prior to surgery, how they came to know about our institution and if they experienced an overall benefit following liposuction. The patients’ pain symptoms were assessed by a numeric rating scale (NRS) ranging from one to ten over twenty items. The subjects included their exact location of pain, sensitivity to touch or pressure, bruising, swelling, extremities’ sensitivity to heat and cold, muscle cramps, heaviness or fatigue of the legs, skin irritation, pruritus and limitations of walking and quality of life, as well as the satisfaction with optical appearance of the legs. The patients’ QOL and level of depression were assessed via an official German version of the WHOQOL-BREF and the PHQ-9. The WHOQOL-BREF measures patient-reported outcomes for overall health. It contains 24 items which are divided into 4 domains to assess every facet of the QOL using a Likert scale from 1 to 5. These health domains include physical health, psychological health, social relationships and the environment. Furthermore, two single “benchmark” items are given to monitor the aspects of general health and overall QOL. The questionnaire is validated for assessing the QOL of patients [34,35].

To screen for the severity of depression, the German version of the PHQ-9 was provided. It consists of nine different items to screen for the risk of developing depression. Participants were asked to provide the frequency of symptom appearance during the last 28 days through a four-point Likert scale (from zero = not at all to four = nearly every day). Five to nine means “minimal symptoms”, ten to fourteen implies “moderate depression” and twenty or more points indicates “severe depression” [36].

Statistical analysis: Following data collection in an individually designed database, a retrospective analysis was performed using IBM SPSS Statistics 27 (IBM, Armonk, NY, USA).

For descriptive analysis, absolute and relative frequencies were computed for the categorical variables. For continuous variables, the mean and standard deviation as well as the median and quartiles were calculated. 

All statistical analysis was of explorative character. Kruskal–Wallis tests were performed to determine statistically relevant differences between the three stages with the dependent variable of the PHQ-9 score before and after the surgery. We used a linear mixed model regression analysis with the random effect being the patient and the fixed effect being the time variable, before and after the surgery. Dependent variables such as the PHQ-9 value were analyzed in the mixed model. We present the effect estimate, the *p*-value and the 95% confidence interval (CI).

## 3. Results

Patients who underwent liposuction in our hospital were analyzed based on questionnaire results. A total number of 56 surgically treated patients were included. Thirty-four of those patients (60.71%) also filled in the questionnaire prior to surgery at their initial presentation before liposuction. Twenty-two (39.28%) completed the questionnaire after liposuction only. An amount of 3831 mL (±1971.08 mL) of pure fat was aspirated on average per procedure and per patient. Patients had a median of 2 (mean = 2.38) liposuctions. In total, 30.36% of patients had only one surgery, 28.57% had two surgeries, 28.57% had three surgeries and only 12.5% had four or more surgeries. These were performed on different anatomical sides. In 57% of cases the lower leg was treated, 23% had their upper leg treated, in 9% of cases surgery was performed on the buttocks and11% were treated on the arms.

### 3.1. Stage of Lipedema

Figure 1 depicts the distribution of lipedema stages for patients at initial presentation at the hospital and the stage they presented with when undergoing surgery. In total, 33.6% of patients prior to surgery initially presented with stage three lipedema. In relation to that, 61.8% presented with a stage three lipedema right before surgery.

Table 2 graphically depicts all variables with their means of assessment and the results of the mixed regression model that will be specifically described further on in the text.

### 3.2. Weight and BMI

Patients presenting for liposuction had a mean age of 40.72 (±12.54) years. Patients prior to liposuction at the primary presentation had a mean body weight of 96.16 kg (±23.11) and a median body weight 93.00 kg (q25 = 79.00, q75 = 110.00). The mean BMI was 33.13 kg/m^2^ (±7.8) and the median BMI was 33.12 kg/m^2^ (q25 = 27.70, q75 = 38.14) across all stages. 

Patients following liposuction had a mean body weight of 93.69 kg (±19.9) and a median body weight of 92.50 kg (q25 = 82.00, q75 = 102.50) as well as a mean BMI of 32.61 kg/m^2^ (±6.69) and median BMI of 32.65 kg/m^2^ (q25 = 27.99, q75 = 35.18). The application of the mixed regression model showed a relevant difference between BMI at initial presentation and after surgery with BMI being 1.65 times higher before the surgery (*p*-value = 0.002, 95% CI 0.67–2.64), as depicted in Table 2. In stage one lipedema, patients had a mean BMI of 26.1 kg/m^2^ (±2.3). The mean BMI of stage two lipedema patients was 29.5 kg/m^2^ (±4.5) and in stage three lipedema, the mean BMI was 34.7 kg/m^2^ (±7.1).

### 3.3. Symptoms in Daily Life

Patients prior to surgery showed a mean of 6.68 (±2.29) and a median of 7.00 (q25 = 5.00, q75 = 8.00) for pain on a numeric rating scale from 1 to 10 in the affected areas. Patients after surgery had a mean for pain in the affected area of 4.29 (±2.16) with a median of 4.00 (q25 = 3.00, q75 = 6.00). Figure 2 shows the decrease in pain on a numeric rating scale from 1 to 10 prior and after the surgery. The mixed model demonstrated 2.67 times higher pain questionnaire scores before surgery compared to after the surgery (*p* = <0.001 95% CI 2.09–3.25), as seen in Table 2.

Pain and pressure sensitivity prior to surgery was assessed with a mean of 7.32 (±2.42) and a median of 8.00 (q25 = 6.00, q75 = 9.00). Patients after surgery demonstrated a mean of 5.46 (±2.73) with a median of 8.00 (q25 = 6.00, q75 = 9.00). In the mixed model, pain and pressure sensitivity was 1.97 times higher prior to surgery (*p* = <0.001 95% CI 1.21–2.71) (Table 2).

The sensation of tension prior to surgery with a mean of 7.49 (±2.19) and a median of 8.00 (q25 = 6.00, q75 = 9.00) can be compared with a mean of 5.50 (±3.01) and a median of 5.00 (q25 = 3.00, q75 = 8.75) after surgery. The mixed model demonstrated 2.17 times higher scores before surgery (*p* = <0.001 95% CI 1.36–2.98). A heavy leg sensation prior to surgery accrued a mean of 8.21 (±1.95) and a median of 9.00 (q25 = 7.00, q75 = 10.00) as opposed to a mean of 5.11 (±3.00) and a median of 5.00 (q25 = 2.25, q75= 8.00) after surgery. In the mixed model, heavy leg sensation was 3.33 times higher prior to surgery (*p* = <0.001 95% CI 2.52–4.13) (Table 2).

The limitations of walking prior to surgery with a mean of 6.45 (±2.78) and median of 7.00 (q25 = 5.00, q75 = 9.00) showed an improvement after surgery with a mean of 4.43 (±2.80) and a median of 5.00 (q25 = 2.00, q75 = 7.00). The mixed model showed 2.26 higher values prior to surgery (*p* = <0.001 95% CI 1.49–3.03). Additionally, the reduction in QOL prior to surgery with a mean of 7.38 (±2.33) and a median of 8.00 (q25 = 6.00, q75 = 9.00) improved to a mean of 5.09 (±2.88) with a median of 5.00 (q25 = 3.00, q75 = 7.00) after surgery. The mixed model showed a 2,9 higher value prior to surgery (*p* = <0.001 95% CI 2.15–3.58). Overall, satisfaction with extremity appearance showed a mean of 9.20 (±1.48) prior to surgery with a median of 10.00 (q25 = 9.00, q75= 10.00) and a mean of 8.38 (±2.80) after surgery with a median of 6.50 (q25 = 4.00, q75= 9.00). The mixed model demonstrated a highly considerable difference between overall satisfaction with the appearance of the extremities and values of the questionnaire were 3.12 times higher before surgery (*p* = <0.001 95% CI 2.40–3.84) (Table 2).

### 3.4. Mental State and PHQ-9

As reflected in Figure 3a,b, the PHQ-9 showed a mean value of 10.84 (±6.38) with a median of 10.00 (q25 = 6.00, q75= 15.00) prior to surgery, which suggests a moderate to severe depressed mood in affected patients. In comparison, after surgery, a mean value of 8.27 (±6.45) and median value of 7.00 (q25 = 3.00, q75 = 11.75) was shown, which suggests a mild depressive mood. The Kruskal–Wallis sample (Figure 3c) showed the differences in depression rates in correlation to the stage of lipedema comparing before and after surgery.

The mixed model demonstrated that the PHQ-9 values were 2.37 times higher before the surgery (*p* = 0.003, 95% CI 0.84–3.89) (Table 2).

### 3.5. Mental State and WHOQOL-BREF 

Figure 4 shows the different WHOQOL domains prior to and after surgery. 

The WHOQOL-BREF mean value for the physical domain prior to surgery was 54.54 (±20.10) with a median value of 57.14 (q25 = 39.28, q75 = 71.43). On the other hand, the WHOQOL-BREF questionnaire after surgery showed a higher mean of 60.33 (±19.98) with a median value of 60.71 (q25 = 46.42, q75 = 78.57) for the physical domain (Figure 4). In the mixed model, scores for the physical domain prior to surgery were 8.85 times lower than after the surgery (*p* value of <0.001, 95% CI −12.84–−4.86) (Table 2).

The psychological domain prior to surgery had a mean of 51.85 (±18.67) with a median value of 54.16 (q25 = 37.50, q75 = 66.66), which was the lowest score obtained. After surgery, the mean was 57.51(±18.31) with a median value 58.33 (q25 = 45.83, q75 = 70.83) (Figure 4). For the mixed model, the values were 4.29 times lower prior surgery (*p*-value of 0.09, 95% CI −9.27–−0.69) (Table 2).

The social domain prior to surgery obtained a mean of 63.72 (±23.05) with a median of 66.67 (q25 = 50.00, q75 = 83.33) compared to a post-surgical mean of 68.42 (±20.23) with a median of 75.00 (q25 = 58.33, q75 = 83.33) (Figure 4). For the mixed model, the values for the social domain were 3.12 times lower prior the surgery (*p*-value= 0.242, 95% CI −8.44–−2.19). (Table 2). The highest score prior to surgery was obtained by the environmental domain with a mean of 71.85 (±16.00) and a median value of 71.87 (q25 = 62.50, q75 = 84.37). However, after surgery it was 74.50 (±16.06) with a median value of 75.00 (q25 = 65.62, q75 = 87.50) (Figure 4). For the mixed model, the values were 3.31 times lower prior to the surgery (*p*-value = 0.084, 95% CI −7.11–0.48) (Table 2).

### 3.6. Occupational Disability

Prior to surgery, 43.9% of patients stated a very severe occupational disability. Five percent were totally disabled to work. Forty-one percent showed a moderate occupational disability. After surgery, 32.1% of patients stated a very severe occupational disability and only 1.8% stated not to be able to work at all. In total, 50% were moderately disabled and 16.1% did not experience any restrictions. In the mixed model, values prior to surgery were 0.37 times higher compared to after the surgery (*p* values = <0.001, 95% CI 0.19–0.55) (Table 2). Sixty-two percent reported to be able to perform more physical activity after surgery.

### 3.7. Post-Operative Complications

Twenty-five percent of patients did not have any post-operative complications at all. The mean pain values after surgery were 5.98 (±2.21). In total, 14% of patients had post-operative pain for up to 7 days, 35.7% for up to 14 days and 50% for more than 14 days. Post-operative swelling lasted for up to 7 days for 7.2% of patients, up to 14 days for 16.1% and more than 14 days for 76.8% of patients. Circulatory problems after surgery lasted up to 7 days for 49.1% of patients, up to 14 days for 45.3% and more than 14 days only for 5.7% of patients.

### 3.8. Patient Satisfaction of the Liposuction

Figure 5 demonstrates that 46% of patients treated by liposuction stated they were very satisfied with the results. Of all the patients, 32.1% were very satisfied, 14.3% were moderately satisfied, 5.4% did not profit at all and 1.8% stated a deterioration of the disease.

There was a positive correlation between high PHQ-9 values and dissatisfaction with appearance. (r = 0.287, *p* = 0.032). By implication, this means higher scores of satisfaction in terms of the appearance of limbs are linked to lower PHQ-9 scores.

### 3.9. Support Groups and Second Opinions

Of the 56 patients, 19.6% sought a support group prior to surgery and 53.6% consulted another physician for a second opinion before the decision of undergoing liposuction for lipedema treatment at our institution. 

## 4. Discussion

This study evaluates the effect of liposuction on the QOL of lipedema patients by patient reported outcome measures (PROMs). This was compared to results obtained prior to the surgery. Based on a multimodal questionnaire with validated scores, such as the PHQ-9 and the WHOQOL-BREF, our results demonstrate a beneficial effect on practically every measured aspect of patients’ physical and psychological wellbeing.

As suggested by the first clinical randomized controlled trial by Podda et al. in 2021 [26], liposuction for lipedema treatment is a promising approach and is either suggested as an additional or alternative treatment to the conservative approach [26]. As previous studies by Wollina et al. [24] and Schmeller et al. [31] demonstrated, liposuction is generally a safe procedure.

There is evidence for surgical treatment efficacy in improving QOL [23,37,38] and overall, previous studies showed positive effects upon social and daily life as well as on general health status [33,38]. Additionally, there is no age limit for the beneficial effects of liposuction for patients [29].

Those findings are endorsed and subsidized by our study, as we included patients between the ages of 18 and 81, who demonstrated an overall statistically relevant improvement after undergoing surgery. As the study of Cobos et al. and Kruppa et al. demonstrated, stage one and two patients showed the highest improvement after liposuction leading to a positive correlation between early-stage surgical treatment and long-term outcomes of the disease [33,39]. These results underline the importance of timely disease diagnosis and treatment.

The findings of a recent article by Baumgartner et al. demonstrate that even in advanced stages there was a benefit of liposuction for patients being treated for lipedema [28]; however, it did not provide substantiated data for stage three patients.

Most patients who underwent surgery at our institution presented with stage two or three lipedema, as shown in Figure 1. Sixty-one percent presented with stage three lipedema. This may suggest a higher demand for surgery in advanced stages due to more severe symptoms and a perceived reduction in QOL. Compared to other studies we evaluated, a strikingly high number of stage three lipedema patients identify liposuction as a safe, effective and complication-low therapeutic option, even in advanced stages.

We compared a relatively large preoperative population with a small number of patients who received liposuction. This can be explained by the fact that in Germany, currently only stage three lipedema is covered by health insurance and other patients have to cover costs out of pocket.

Especially in congruence with Cobos et al. and Kruppa et al. [33,39], this should raise awareness to the fact that many patients do not have access to this type of therapy, despite the need for it. Potentially, this contributes to disease progression with all its consequences for the individual patient.

Our study results demonstrated an overall improvement of all numerical scales of daily symptoms after surgery, such as pain, bruising, sensitivity to pressure and cosmetic outcome. This correlates with previous publications of Dadras et al. and Münch et al. as well as long-term outcomes assessed by Baumgartner et al. [23,28,40]. Especially for the pain symptoms before and after surgery, our results supported an improvement after liposuction.

Comparing the psychological assessment of the questionnaire, we noted a change from moderate to severe depression prior to surgery to a mild depressive mood after liposuction, implying decreased depression levels after surgery. We can see a difference after surgery in PHQ-9 values being lower. While the study of Bertsch et. Al. demonstrated the mental health effects of the disease [13], a study by Papadopulos et al. performed its own liposuction-specific questionnaire on 38 patients, which also showed an overall improvement in QOL and psychological wellbeing [41]. While these studies use single items to evaluate the potential benefits, our study also verified the impact on the psychological health of the affected patients using a multimodal questionnaire with the validated PHQ-9.

The study of Bertsch et al. recommended psychologically affiliated treatment for the surgical concept [13]. As PHQ-9 scores after surgery were still reflecting mild depression, psychological concomitant therapy may be a promising approach.

Concerning the WHOQOL-BREF questionnaire’s physical domain, higher values were given for patients after surgery, which suggests not only a subjective improvement after liposuction but also showed a statistically relevant difference between the group prior to surgery and the group after surgery. This validated questionnaire shows a clear improvement in physical symptoms after liposuction. Compared to other studies, this validated WHOQOL-BREF questionnaire gives especially substantial results.

In conclusion, an improvement in each separate domain after surgery was demonstrated. However, only the physical domain of the questionnaire shows statistical relevance. Nevertheless, the results insinuate a similar direction of improvement in all domains by demonstrating higher numbers after surgery. 

Additionally, we were able to denote a difference in the occupational disability of patients prior to and after surgery with lower levels of occupational disability after undergoing surgery. This again emphasizes the importance and economic relevance of liposuction regarding the professional life of patients. Patients in our study also stated being able to perform more physical activity in their private life, such as going for walks or runs and swimming, which can contribute to psychological wellbeing and perceived improvement of quality of life.

As 92.8% of patients who underwent liposuction stated to be satisfied with their surgical results, liposuction can have a beneficial effect both subjectively and objectively. 

It is not only important to ease physical and psychological discomfort but also to prevent secondary complications resulting from the disease, including lipo/lymphedema, joint deformities, skin infections and morbid obesity [42,43]. Liposuction does not cure lipedema; however, it may improve the quality of life of patients by relieving disabling symptoms, minimizing progression and preventing further complications.

Another important fact to underline is the change in weight and thereby BMI. Even though there is a relevant difference between the patient group prior to and after surgery, we would like to point out that liposuction has the primary goal of improving QOL and pain levels, independent of weight as a single factor. 

Liposuction does not automatically change a patients’ BMI and alleviate all associated clinical signs and symptoms; self-evidently other conservative modalities need to be applied. Physical activity, as well as a healthy lifestyle and diet, remain crucial domains for lipedema patients to prevent them from relapses. Suggestively, BMI may not be a valid factor for monitoring treatment outcomes of lipedema patients. Since we found a correlation between the satisfaction of appearance and PHQ-9 values, this factor seems to play an important role in patients’ wellbeing. The more satisfied patients entailed those with their affected limbs’ appearance accruing lower PHQ-9 scores and depression. However, the interplay of satisfaction with appearance, the ability to perform more physical exercise, the reduction of daily symptoms and the ability to evolve inprofessional life contribute to the overall improvement of QOL of lipedema patients after liposuction.

We suggest a standardized and multimodal questionnaire, as we performed in our study, which contains validated questionnaires such as the WHOQOL-BREF and the PHQ-9 score to better assess treatment outcomes. This could make the various studies more comparable and allow for uniform follow-up after liposuction and more substantiated conclusions.

The socioeconomics of this disease is extensive and this study should additionally spread awareness about the necessary measures for optimal treatment. This is referring to both socioeconomic factors of the hospitals and patients’ QOL. 

As this study suggested, patients’ characteristic complaints decreased after liposuction, improving the QOL of patients not only physically but also psychologically. 

As previous studies insinuated, our study supported the understanding that liposuction has general improving effects on QOL, both in private and professional life. This study illustrated a step forward by including a large number of stage three patients and identifying liposuction as a safe and effective form of therapy, as well as consolidating beneficial effects by validated PHQ-9 and WHOQOL-BREF. 

These results encourage the proposal of liposuction as a standardized and easily accessible treatment option for patients suffering from lipedema.

Concerning the limitations of our study, since an explorative analysis was performed, *p* values should be interpreted in correlation with effect estimates and we may speak of statistical relevance or pertinence. A recall bias cannot be excluded. We compared a relatively large preoperative population with a small number of patients who received liposuction. Hence, assessment of the long-term outcome of liposuction with a larger number of patients remains to be performed within a longer follow-up period. Moreover, matching lipedema patients and further comparing conservative measures only with the surgical approach and its effects on satisfaction and QOL would be an interesting approach for a follow-up study. 

## 5. Conclusions

Lipedema remains a complex disease associated with a high number of individuals as well as a socio-economic burden. Liposuction represents a safe and effective procedure for lipedema treatment in all stages of the disease, alleviating symptoms and improving QOL. Liposuction, as the standard of care, may contribute to the impending better control and treatment outcomes of the disease.

## Figures and Tables

**Figure 1 healthcare-11-02020-f001:**
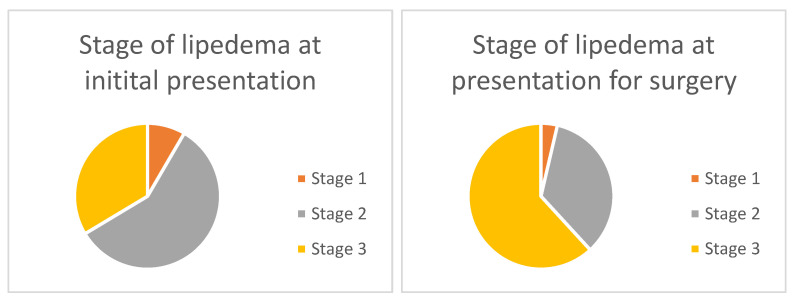
Lipedema stages at initial presentation and when undergoing surgery. At initial presentation, 8.6% of patients presented with stage one lipedema, 57.8% with stage two and 33.6% with stage three lipedema. At presentation for surgery, 3.65% of patients presented with stage one, 34.5% with stage two and 61.8% with stage three lipedema.

**Figure 2 healthcare-11-02020-f002:**
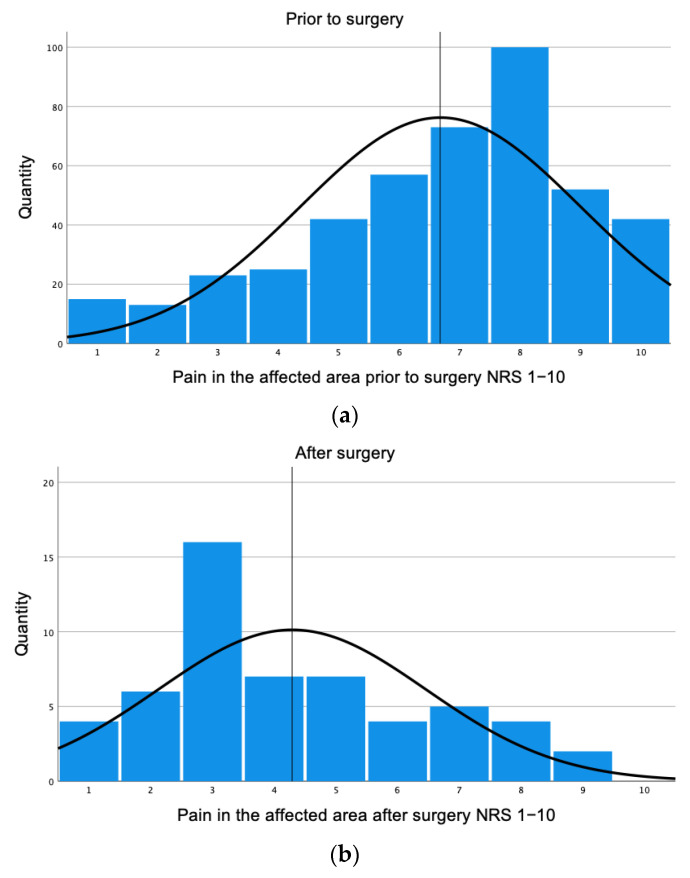
Pain in affected areas prior and after surgery. (**a**) Prior to surgery, patients showed a mean of 6.68 (±2.29). (**b**) Patients after surgery showed a mean of 4.29 (±2.16).

**Figure 3 healthcare-11-02020-f003:**
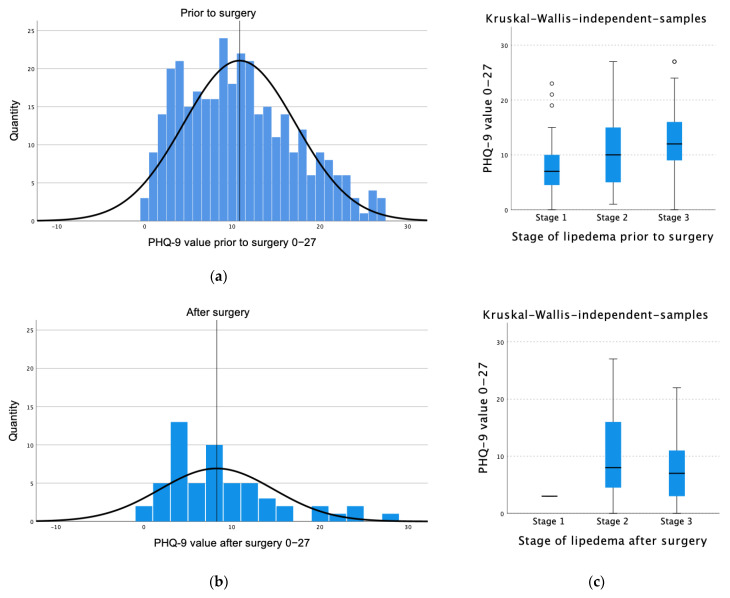
PHQ-9 value prior to and after surgery. (**a**) Prior to surgery, a mean value of 10.84 (±6.39) insinuates moderate to severe depression. (**b**) After surgery, the mean value is 8.27 (±6.45), indicating a mild depressive mood. (**c**) Differences in depression rates in correlation to the stages of lipedema comparing before and after surgery.

**Figure 4 healthcare-11-02020-f004:**
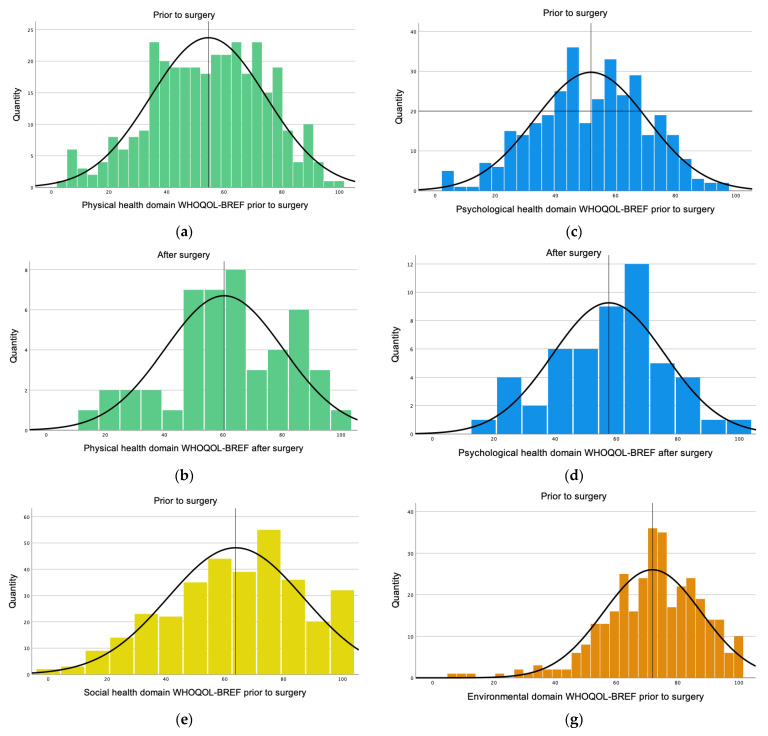
WHOQOL-BREF domains. (**a**) The mean value of the physical domain was 54.54 (±20.10) prior to surgery. (**b**) After surgery, the mean value for the physical domain was 60.33 (±19.98). (**c**) The psychological domain prior to surgery had a mean of 51.85 (±18.67). (**d**) After surgery, the mean for the psychological domain was 57.51(±18.31). (**e**) The social domain prior to surgery obtained a mean of 63.72 (±23.05). (**f**) After surgery, the mean of the social domain was 68.42 (±20.23). (**g**) Prior to surgery, the environmental domain had a mean of 71.85 (±16.00). (**h**) After surgery, the mean for the environmental domain was 74.50 (±16.06).

**Figure 5 healthcare-11-02020-f005:**
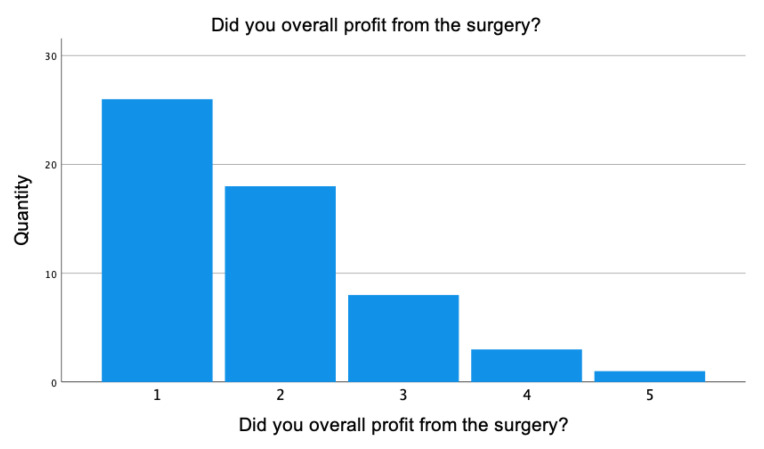
Patient satisfaction after surgery. 1 = very satisfied, 2 = satisfied, 3 = moderately satisfied, 4 = not at all, 5 = deterioration. The mean value of 56 patients was 2.00 (±0.98).

**Table 1 healthcare-11-02020-t001:** Lipedema diagnostic criteria [2].

Almost exclusive occurrence in women	Easy bruising
Bilateral and symmetrical manifestation with minimal involvement of the feet	Arms are affected 30% of the time *
Persistent enlargement after elevation of the extremities or weight loss	Hypothermia of the skin *
Minimal pitting edema	Swelling worsens with orthostasis in summer *
Negative Kaposi–Stemmer sign	Unaffected by caloric restriction *
Pain, tenderness on pressure	Telangiectasias *

* Added by Herbst [3,6].

**Table 2 healthcare-11-02020-t002:** Linear mixed model with variable, effect estimate prior to surgery, *p*-value and 95% CI.

Variable	Means of Assessment	Estimate Value	*p*-Value	95% CI
BMI	Weight, Height in cm	1.65	0.002	0.67–2.64
Pain in affected area	NRS from one to ten	2.67	<0.001	2.09–3.25
Pain and pressure sensitivity	NRS from one to ten	1.97	<0.001	1.21–2.71
Sensation of tension	NRS from one to ten	2.17	<0.001	1.36–2.98
Heavy leg sensation	NRS from one to ten	3.33	<0.001	2.52–4.13
Limitation of walking	NRS from one to ten	2.26	<0.001	1.49–3.03
Reduction in QOL	NRS from one to ten	2.9	<0.001	2.15–3.58
Overall satisfaction with the appearance of the extremities	NRS from one to ten	3.12	<0.001	2.40–3.84
PHQ-9 values	PHQ-questionnaire for depression	2.37	0.003	0.84–3.89
Physical domain of WHOQOL-BREF	WHOQOL-BREF physical health with Likert scale from one to five.	−8.85	<0.001	−12.84–−4.86
Psychological domain of WHOQOL-BREF	WHOQOL-BREF psychological health with Likert scale from one to five	−4.29	0.09	9.27–−0.69
Social domain	WHOQOL-BREF social health with Likert scale from one to five.	−3.12	0.242	−8.44–2.19
Environmental domain	WHOQOL-BREF environmental health with Likert scale from one to five	−3.31	0.084	−7.11–0.48
Occupational disability	Effects on ability to work	0.37	<0.001	0.19–0.55

## Data Availability

The data presented in this study are available on request from the corresponding author.

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
