# Peer review of "Patient-Reported Outcomes of Liposuction for Lipedema Treatment"

_healthcare, 2023, doi:10.3390/healthcare11142020_

Round 1
Reviewer 1 Report
Kirstein and colleagues present work following patient QOL before and after liposuction surgery for lipedema in their submitted work. Various important QOL aspects are measured including mental health and ability to work. In total, the work is important for the field of lipedema. There are some concerns with the presentation of the data and its conclusions.
Could the authors please provide a breakdown of patient information by lipedema stage (for example, for BMI, WHR) and provide more information as to the number of surgeries, anatomical site of surgeries, etc.? It may be important to then examine these details against some outcomes.
Section 3.8 reports that not all patients were satisfied with their procedures and also a range of satisfactions were indicated. It would be both interesting and important to correlate these responses with the other measures graphed. It is presumable that a patient who was not pleased would also have residual QOL issues. Likewise, a patient highly satisfied would very much demonstrate great QOL? Is satisfaction linked to any parameters specifically? Maybe not even QOL, but merely cosmetic limb volume reduction or scarring?
While asked, there is not further discussion in the Results as to whether patients were on any anti-depressant medications, sought support groups, etc.
Table 1 is generically 'lipedema'. Are these the official diagnositic criteria utilized by the treating physicians?
The authors make statements that liposuction could or should be standard of care, but no comparisons are made between outcomes of patients who do no exhibit lipedema (who have liposuction; and so are satisfaction and QOL outcomes equal) or of those lipedema patients who elected to not have surgery and instead sought other remedies. This MUST be described in the text and conclusions tempered.
Minor:
It is not clear in the methods if all patients are being treated for lipedema.
If pain goes away, does the patient still have lipedema or are they cured?
The writing is imperfect and some odd vocabulary choices are utilized.
Reviewer 2 Report
see attached file for all comments about the article

Round 2
Reviewer 1 Report
The authors have been very responsive to the previous concerns and the work is acceptable.
Reviewer 2 Report
great comments provided with the relevant corrections